# The Integrity of α-β-α Sandwich Conformation Is Essential for a Novel Adjuvant TFPR1 to Maintain Its Adjuvanticity

**DOI:** 10.3390/biom9120869

**Published:** 2019-12-12

**Authors:** Qiao Li, Xiuzhe Ning, Yuepeng Wang, Qing Zhu, Yan Guo, Hao Li, Yusen Zhou, Zhihua Kou

**Affiliations:** 1Beijing Institute of Microbiology and Epidemiology, Anhui Medical University, Hefei 230032, China; 18500797564@163.com (Q.L.); wyp904850758@outlook.com (Y.W.); 2State Key Laboratory of Pathogen and Biosecurity, Beijing Institute of Microbiology and Epidemiology, Beijing 100071, China; teamo515@163.com (X.N.); zq1010863746@163.com (Q.Z.); muhan0425@126.com (Y.G.); lihao88663239@126.com (H.L.); zhouysbj01@163.com (Y.Z.)

**Keywords:** Pathogenesis-relatedprotein1 (PR-1), conformational structure, adjuvant, dendritic cells (DCs), peptide antigens, B cell epitope

## Abstract

TFPR1 is a novel peptide vaccine adjuvant we recently discovered. To define the structural basis and optimize its application as an adjuvant, we designed three different truncated fragments that have removed dominant B epitopes on TFPR1, and evaluated their capacity to activate bone marrow-derived dendritic cells and their adjuvanticity. Results demonstrated that the integrity of an α-β-α sandwich conformation is essential for TFPR1 to maintain its immunologic activity and adjuvanticity. We obtained a functional truncated fragment TFPR-ta ranging from 40–168 aa of triflin that has similar adjuvanticity as TFPR1 but with 2-log fold lower immunogenicity. These results demonstrated a novel approach to evaluate and improve the activity of protein-based vaccine adjuvant.

## 1. Introduction

Adjuvants have proven to be vital components in vaccines [1,2,3], but a very small number of adjuvants that have been licensed for human use are not optimal for many various types of vaccines. For example, aluminum salts, that are safe and widely applied to the formulation of human vaccines, have little capability to enhance the immunogenicity of peptide antigens or effectively augment cellular immune responses [4,5,6,7,8]. Oil-in-water emulsions such as MF59, AS03, and AF03, or adjuvant combination AS04 (monophosphoryl lipid A plus aluminum salts) are also less effective for peptide antigens, and emulsion may increase the risk of adverse reactions [9,10,11,12]. Therefore, attempts to develop novel adjuvants that are capable of inducing both humoral and cellular immune responses safely and suitable for novel vaccines including peptide vaccines, have been made and some progress achieved. Immunostimulatory adjuvants, such as TLR4 ligand MPLA [13,14], TLR5 ligand Flagellin [15,16], and TLR9 ligand CpG-ODN [17,18] have been demonstrated to be capable of stimulating innate immune responses directly and efficiently [19,20], that play pivotal roles in modulating the subsequent induction of adaptive immune responses.

TFPR1 is a novel adjuvant derived from a non-toxic protein, triflin, which belongs to the CAP superfamily—cysteine-rich secretory proteins (CRISPs), antigen 5 (Ag5) and pathogenesis-related protein 1 (PR-1); it comprises a cysteine-rich domain (CRD) at the C-terminus and a PR-1 domain at the N-terminus [21,22,23]. Accordingly, TFPR1 retains the conserved core PR-1 domain and the CAP signature motifs, just as the other members in the CAP superfamily [24]. Our previous studies indicated that recombinant protein TFPR1 augmented the immunogenicity of protein (OVA and recombinant HBsAg) and peptide (HIV-1 pep5 envelope) antigens, and induced Th1-biased antibody- and cell-mediated immune responses. Its adjuvant activity may be related to its capacity of activating antigen-processing cells, such as dendritic cells [25], B cells and macrophages, and promoting the secretion of Th1-biased proinflammatory and immunoregulatory cytokines [26]. However, as a protein adjuvant, TFPR1 has relatively strong immunogenicity [27], the structural basis of its function is not yet defined.

In this study, we not only confirmed that TFPR1 acts as an effective adjuvant for peptide antigen, but also identified its minimal functional region. By rationally design functional fragment with minimum immunogenicity, through deleting B cell dominant epitopes while retaining its main structure and motifs, and expressed these truncated forms in *E.coli* protein expression system, we obtained a functional truncated fragment TFPR-ta ranging from 40–168 aa of triflin that has similar adjuvanticity as TFPR1 but with 2-log fold lower immunogenicity. Our results suggest that the integrity of the typical α-β-α sandwich conformational structure is essential for TFPR1 protein to maintain its adjuvanticity, and α1-helix and β4-fold cannot be deleted at the same time.

## 2. Materials and Methods

### 2.1. Mice

BALB/c mice (Beijing Experimental Animal Center, Beijing, China) were used for the mouse experiments. This study was carried out in strict accordance with the recommendations in the *Guide for the Care and Use of Laboratory Animals* of the Ministry of Science and Technology of People’s Republic of China. Protocol BIME 2014-16 was approved by the Institutional Animal Care and Use Committees of Beijing Institute of Microbiology and Epidemiology.

### 2.2. Evaluation of Adjuvanticity of TFPR1 for the Peptide Antigen HIV-1 CBD

Adjuvanticity of TFPR1 was evaluated in female 6- to 8-week-old BALB/c mice using the synthetic peptide antigen HIV-1 CBD (SLEVIWNNMTWMEWEREIDN), which was derived from the caveolin-binding domain of HIV-1 gp41 [28,29]. Mice (*n* = 6 per group) were immunized intramuscularly with aqueous mixture containing 20 μg of HIV-1 CBD and 10 μg of recombinant TFPR1, or HIV-1 CBD adsorbed to alum (InvivoGen, volume ratio of OVA to alum is 1:9), HIV-1 CBD alone, or endotoxin-free PBS (GIBCO, Grand Island, NY, USA), and boosted once three weeks later. Serum was collected before immunization and on the second week after the final immunization to detect specific anti-HIV-1 CBD antibodies IgG, IgG1 and IgG2a by ELISA, HIV-1 CBD at a final concentration of 1 μg/mL was used as the capture antigen. The protocol is similar to that described previously [30]. The antibody titer was defined as the reciprocal of the largest dilution of serum which OD_450_ value was greater than average value of the negative control (before immunization) plus two-fold of the standard deviation (SD).

### 2.3. Analysis and Identification of B Cell Epitopes on TFPR1

The B cell epitopes on TFPR1 were predicted using IEDB online software (http://tools.iedb.org/main/bcell/), and the structural and functional motifs of TFPR1 were analyzed by the tool “secondary structure consensus prediction” of NPS (http://npsa-prabi.ibcp.fr) and according to the reference [24]; then, the peptides which were the predicted B cell epitope and/or key motifs of TFPR1 were synthesized (Genscript, Nanjing, China), and used as coating antigens in ELISA for further epitope identification; serum was collected from the TFPR1- or TFPR1/Alum-immunized BALB/c mice. The immunization was performed as the following: 6- to 8-week-old female BALB/c mice were immunized intramuscularly with 10 μg of TFPR1 alone or TFPR1 plus alum, and endotoxin-free PBS alone was used as negative control. Serum specific antibody IgG to each peptide were measured by ELISA, the 96-well plate was precoated with each synthetic peptide (2 μg/mL) separately, and TFPR1 was used as positive control, the serum was diluted 1/20 and added into each well. Data was presented as the OD_450_ value, the epitope was defined as positive according to one-way ANOVA analysis compared with negative control.

### 2.4. Expression, Purification, and Structure Prediction of Different Truncated Fragments of TFPR1

As described previously [26,27,28,29,30], the truncated fragments of TFPR1 were designed and inserted into the pQE-30 vector (containing 6 × His-tag coding sequence), and expressed in E.coli. Because the recombinant proteins were expressed in the form of inclusion bodies, after the proteins were purified by Ni-NTA chromatography and dissolved in 8 M urea, the soluble denatured proteins were refolded by gradient dialysis in 6 M, 4 M and 2 M urea in tris-glycine buffer, followed by Laemmli buffer and PBS. The expression of recombinant truncated proteins was analyzed by SDS-PAGE and confirmed by Western blot with mouse sera containing anti-TFPR1 antibody [30]. Finally, the purified proteins were treated with a lipopolysaccharide (LPS)-removing gel (Detoxigel™, Pierce Biotechnology, Rockford, IL, USA) to remove the contaminating LPS. The endotoxin content of the purified proteins was then tested using the Limulus amebocyte lysate (LAL) assay, and the proteins which tested negative were used for the following experiments.

The conformational structures of TFPR1 and its truncated fragments were predicted by online software SWISS-MODEL (http://swissmodel.expasy.org/), difference between the full-length protein TFPR1 and its truncated fragments was compared. The secondary structure of each protein was analyzed using Circular Dichroism [31,32] (Chirascan, Applied Photophysics Ltd, England), and analyzed using CDNN software (Gerald Böhm, Magdeburg, England).

### 2.5. Adjuvanticity Evaluation of Each Truncated Fragment of TFPR1 Using Mouse Bone Marrow–Derived Dendritic Cells in Vitro

Bone marrow cells were collected from 4- to 5-week-old healthy male BALB/c mice under aseptic conditions, and were cultured at a density of 2.5 million per mL in 96-well plates with complete RPMI 1640 (GIBCO, Grand Island, NY, USA) containing rmGM-CSF (final concentration, 10 ng/mL) and rmIL-4 (final concentration, 10 ng/mL) at 37 °C in an atmosphere of 5% CO_2_ [25,33,34]. Cells in triplicate were induced to differentiate in each well for six days, and then were incubated with the truncated fragments of TFPR1 for the following experiments.

Purified truncated fragments of TFPR1 (final concentration, 10 μg/mL) were added to bone marrow-derived dendritic cells, then incubated for 24 h, and TFPR1 was used as a positive control. Then, culture supernatants were harvested, and the levels of IL-6, IL-8, IL-12, and TNF-α were measured using specific ELISA kits (Neobioscience, Shenzhen, China), according to the manufacturer’s recommendations. Cytokine concentration was read from the standard curves and expressed as pg/mL.

### 2.6. Adjuvanticity Evaluation of Each Truncated Fragment of TFPR1 Using the Model Antigen OVA In Vivo

The adjuvanticity of truncated fragments of TFPR1 were evaluated in BALB/c mice immunized with OVA [30]. As described above, 6- to 8-week-old female BALB/c mice (n = 6 per group) were intramuscularly injected with OVA (10 μg) formulated with each truncated fragment of TFPR1 (10 μg), and boosted once with the same antigens and adjuvants three weeks later. OVA alone or endotoxin-free PBS were used as controls. Serum was collected to measure OVA-specific antibodies, IgG, IgG1, and IgG2a by ELISA. Meanwhile, antibody against TFPR1 itself in serum collected on the 2nd week after final immunization was detected to evaluate the immunogenicity of each truncated fragments of TFPR1, and specific anti-TFPR1 antibodies IgG, IgG1 and IgG2a were measured by ELISA. OVA or TFPR1 at a final concentration of 1 μg/mL were used as the capture antigen as described previously. The antibody titer was defined as the reciprocal of the largest dilution of serum which OD450 value was greater than average value of the negative control (before immunization) plus two-fold of the standard deviation.

### 2.7. Statistical Analysis

Statistical analysis was performed using GraphPad Prism, version 5.0 (GraphPad Software, San Diego, CA, USA). The means or geometric means from multiple groups were compared using one-way analysis of variance (ANOVA). A *p* value of less than 0.05 was considered statistically significant.

## 3. Results

### 3.1. TFPR1 Acts as an Effective Adjuvant for Peptide Antigen HIV-1 CBD

To verify whether purified TFPR1 has the capacity to enhance the immunogenicity of B cell epitope-based peptide antigen, HIV-1 CBD [28,29], a 20 aa HIV-1 gp41 peptide, was synthesized and used as immunogen. BALB/c mice were immunized intramuscularly with HIV-1 CBD in the presence or absence of TFPR1, and alum was used as an adjuvant control. ELISA results showed that TFPR1 stimulated higher levels of CBD-specific IgG antibody than alum (*p* < 0.05) (Figure 1A), and TFPR1 induced balanced Th2-associated IgG1 (*p* < 0.01 vs. CBD) (Figure 1B) and Th1-associated IgG2a (*p* < 0.01 vs. CBD) (Figure 1C) antibody responses; in contrast, alum induced 10 fold more IgG1 than IgG2a antibody response. These results suggested that TFPR1 can act as an effective adjuvant for the peptide antigen. However, our previous studies showed that TFPR1 can also induced strong antibody responses to itself in the absence of adjuvant in mice [27], and thus modulating its immunogenicity but retaining its adjuvanticity may be important for its use as an effective adjuvant. 

### 3.2. Prediction and Identification of Dominant B cell Epitopes on TFPR1

We next examined whether it was possible to retain the key functional structure of TFPR1 but reduce its immunogenicity. Nine potential linear B cell epitopes on TFPR1 were predicted using the online software IEDB, six epitopes with high (No. 2, 3, 4, and 6) or intermediate high scores (No. 1 and 5) (Figure 2A and Table 1), among which two continuous epitopes located on Aa 31–35 (No.1) and Aa 45–53 (No.2) form the α1-helix of TFPR1. According to the predicted epitopes and secondary structure and functional fragments [24], we synthesized nine peptides. (named P1–P9) that represent key functional motifs or predicted B cell epitopes (Figure 2B and Table 2). P1 located on aa 31–48 is part of the α1-helix and contains two predicted strong epitopes (No.1 and No.2). These synthetic peptides were tested experimentally.

These predicted B cell epitopes were then tested against TFPR1-immunized mouse sera using ELISA. Strong P1-specific antibody response (*p* < 0.001 vs. NC), and weaker P3- and P6-specific IgG antibody response (Figure 3A) were detected in mice immunized with TFPR1 alone; and high levels of P1- (*p* < 0.001 vs. NC), P3- (*p* < 0.05 vs. NC) and P6-specific IgG antibodies (*p* < 0.05 vs. NC) were detected in mice immunized with TFPR1 plus adjuvant alum (Figure 3B). These results indicated that not all the predicted epitopes with high scores are proven to be real epitopes in BALB/c mice, such as P2 containing epitope #2 with score 1.06, P5 containing epitope #5 with score 0.741, but the predicted weak epitopes with low scores or negative ones were proven to be not epitopes, such as P8 and P9, P4 and P7. Strong epitope P1 (Aa 31 to Aa 48 comprising α1-helix), and two sub-dominant B cell epitopes P3 (aa 57 to aa 71 form the β1-fold and part of α2-helix) and P6 (aa 103 to aa 117, forming α3-helix and containing predicted epitopes No.5 and 6), are verified.

### 3.3. TFPR-T, the Core PR-1 Domain of TFPR1 in Absence of Partial α1-Helix and β4-fold, Has Little Adjuvant Activity In Vivo and In Vitro

Based on the above results, we designed a truncated fragment ranging from 40 to 156 Aa of TFPR1, named TFPR-T, which lacks the N-terminal partial α1-helix (part P1, Aa 31–39) and C-terminal β4-fold (P9, Aa 158–172) (Figure 4A). A 13.2 kDa recombinant protein TFPR-T was obtained in the native form after purification and recovery by dialysis, as verified by SDS-PAGE (Figure 4B), circular dichroism (CD)-spectrum (Appendix A) and Western blotting using anti-TFPR1 prepared in our laboratory (Figure 4C). The adjuvanticity of TFPR-T was tested in BALB/c mice using OVA as immunogen. Results showed that in comparison to TFPR1, TFPR-T induced lower levels of OVA-specific antibodies of IgG (Figure 4D), IgG1 (Figure 4E), and IgG2a isotypes (Figure 4F), indicating that the adjuvanticity of TFPR-T reduced significantly.

Because TFPR1 can efficiently activate dendritic cells to produce high levels of proinflammatory cytokines [25], we then analyzed whether TFPR-T also reduced its ability to stimulate dendritic cells to secrete cytokines. Bone marrow-derived DCs were incubated with either TFPR-T or TFPR1 for 24 h, and ELISA was performed to compare their cytokine secretion. TFPR-T hardly induced any cytokines other than IL-8 (2528 pg/mL, *p* < 0.05 vs. PBS), and significantly lower levels of IL-6, IL-12 and TNF-α than TFPR1 (Figure 5). These results indicate that the reduced adjuvanticity of TFPR-T is linked to ineffective activation of dendritic cells to secrete cytokines.

### 3.4. Analysis, Design and Expression of Different Fragments Containing Either α1-Helix (TFPR-hd) or β4-Fold (TFPR-ta)

To further explore which domain features, P1 which contains α1-helix or P9 which contains β4-fold, is necessary for TFPR1 to maintain its immunologic activity and adjuvanticity, we next analyzed the conformational structure of TFPR-T using online software SWISS-MODEL. To our surprise, the deficiency of partial α1-helix and β4-fold at the same time led to significant conformation disruption of the typical α-β-α sandwich structure of PR-1 family, not only α1-helix and β4-fold disappeared, but also β2- and β3-fold were lost (Figure 6D), suggesting that disruption of the typical α-β-α sandwich structure may be responsible for the lack of adjuvanticity of TFPR-T. Then, we designed a 126-Aa fragment (ranging from 31–156 Aa, named TFPR-hd) which complemented the α1-helix at the N-terminus, and a 129 Aa fragment (ranging from 40–168 Aa, named TFPR-ta) which complemented the β4-fold at the C-terminus of TFPR-T (Figure 6A). The homology modeling by SWISS-MODEL showed that removal of β4-fold on TFPR1 (TFPR-hd) also led to the loss of β2 and β3-fold at the C-terminus, but due to existence of the intact α1-helix, TFPR-hd still retained relative intact α-β-α structure; while, missing of partial α1-helix only at N-terminal of TFPR1 (TFPR-ta) led to no conformational change, thus, TFPR-ta maintained the integrity of α-β-α sandwich structure to the greatest extent with only a partial α1-helix at the N-terminus (Figure 6D), suggesting that these two fragments might have retained the immune activity and adjuvant effect. 

To test whether the modeling analyses are correct, recombinant TFPR-hd and TFPR-ta were expressed in *Escherichia coli*, and proteins in native form were obtained by dialysis. Western blotting showed that both TFPR-hd and TFPR-ta were recognized specifically with mouse sera containing anti-TFPR1 antibody. Notably, although the intensities of TFPR-hd and TFPR-ta bands were similar on SDS-PAGE (Figure 6B), the intensity of TFPR-ta band was much weaker than that of TFPR-hd and TFPR1 on /B (Figure 6C), indicating that immunogenicity of TFPR-ta might be significantly decreased due to the loss of dominant B epitope (P1) at the N-terminus. Also, the CD-spectrum analysis preliminarily proved prediction of secondary structures of the four proteins by SWISS-MODEL (see detailed in Appendix A).

### 3.5. Both TFPR-Hd and TFPR-Ta Activate Dendritic Cells to Produce Cytokines Effectively

The adjuvanticity of TFPR-hd and TFPR-ta were evaluated with dendritic cells in vitro and the model antigen OVA in vivo. TFPR-hd and TFPR-ta were used to stimulate dendritic cells for 24 h to examine their potential adjuvanticity. Similar to TFPR1, both TFPR-hd and TFPR-ta elicited significant levels of various cytokines, including IL-6 (3695 and 3700 pg/mL, *p* < 0.05 vs. PBS) (Figure 7A), IL-8 (6203 and 6238 pg/mL, *p* < 0.001 vs. PBS) (Figure 7B), IL-12 (3223 and 2644 pg/mL, *p* < 0.001 vs. PBS) (Figure 7C) and TNF-α (3552 and 2554 pg/mL, *p* < 0.001 vs. PBS) (Figure 7D). These results suggested that both TFPR-hd and TFPR-ta are effective in stimulating dendritic cells to secrete cytokines, indicating that relative integrity of α-β-α sandwich structure is essential to maintain the immune regulatory activity of the TFPR-1 protein.

### 3.6. TFPR-Hd and TFPR-Ta Retain the Adjuvanticity of TFPR1 for the Model Antigen OVA

Having shown that TFPR-hd and TFPR-ta were able to activate dendritic cells, we subsequently assessed their adjuvant effect on OVA in mice. As with TFPR1, both TFPR-hd and TFPR-ta elicited strong IgG (*p* < 0.001) (Figure 8A), and subclass IgG1 (*p* < 0.001) (Figure. 8B) and IgG2a (*p* < 0.001) (Figure 8C) antibody responses to OVA. Meanwhile, analysis of the immunogenicity of these proteins showed that different from TFPR1 and TFPR-hd that induced high levels of TFPR1-specific IgG, and subclass IgG1 and IgG2a antibodies, TFPR-ta (as TFPR-T did) induced about 2-log fold lower anti-TFPR1 antibody than TFPR1 and TFPR-hd (Figure 8D, E and F). Taken together, both TFPR-hd and TFPR-ta retained the adjuvanticity of the full-length protein TFPR1 in vitro and in vivo, demonstrating that the relative integrity of α-β-α sandwich structure is vital to its adjuvant effect. The high immune regulatory capability and low immunogenicity of TFPR-ta makes it more suitable for development as an adjuvant product.

## 4. Discussion

As a key component of many vaccine formulations, an adjuvant enhances and shapes antigen-specific immune responses [35,36,37]. TFPR1 was discovered as a novel non-toxin protein adjuvant previously [26,27]. We showed it is capable of significantly augmenting the immunogenicity of various antigens, inducing Th1-biased antibody- and cell-mediated immune responses. In this follow-up study, we further characterized the structural feature that is critical for the adjuvanticity of TFPR1 using a B cell epitope-based peptide antigen, HIV-1 CBD, and defined a functional region with low immunogenicity. 

Compared to another novel adjuvant Ov-ASP-1 [38] and its functional truncates ASPPR [30], which has a homological structure as TFPR1, TFPR1 might be superior in terms of application due to its lower effective dose [26]. In addition, as a protein, TFPR1 has relative strong immunogenicity itself [27], which is consistent with other protein adjuvants, such as flagellin [39], Ov-ASP-1, etc., although the immunogenicity may not affect their adjuvant effect as proved in several studies [40,41]. However, the strong immunogenicity of TFPR1 might affect its further application in the future, for example, there might be competent with vaccine antigens. Therefore, finding the functional domain of TFPR1, thus obtaining a minimum functional fragment with lower immunogenicity is an important issue.

Deletion of B cell dominant epitopes is an effective way to reduce the immunogenicity of proteins [42,43]; therefore, after screening and identifying B cell epitope on TFPR1, we designed TFPR-T, which lacks of partial α1-helix (deleting 9 Aa at the N-terminus) and β4-fold (19 Aa at the C-terminus) on TFPR1 in order to obtain a minimum functional fragment with low immunogenicity, and was expressed successfully in E.coli. As expected, the immunogenicity of TFPR-T was significantly reduced, but unexpectedly, TFPR-T lost ability to augment significant specific antibodies in mice, and almost lost its ability to activate mouse dendritic cells to generate proinflammatory cytokines in vitro, suggesting that adjuvanticity loss of TFPR-T is relevant to its disability to activate dendritic cells. Noticeably, SWISS-MODEL analysis and CD-spectrum indicated that the absence of a α1-helix and a β4-fold at the same time led to completely disruption of the typical α-β-α sandwich structure of TFPR1, suggesting the potentially close relationship between the intact α-β-α sandwich structure and dendritic cell activation and adjuvanticity of TFPR1. Dendritic cells are the most powerful antigen presenting cells, and many adjuvants act through activating dendritic cells [44,45,46,47,48]. In our previous study, TFPR1 has been proved to be capable to activate dendritic cells [25]. Considering findings above, we conclude that the capability of TFPR1 to activate dendritic cells in vitro is closely related to the adjuvanticity in vivo. Therefore, we may predict the adjuvanticity of a protein adjuvant which mechanism is mainly through activating dendritic cells using dendritic cells stimulating assay in vitro before we conduct mice experiments; and it is necessary to investigate which domain, α1-helix and β4-fold, play important role of adjuvanticity for TFPR1. 

Then, we designed another two truncated fragments, TFPR-hd which complemented α1-helix at the N-terminus, and TFPR-ta which complemented β4-fold at the C-terminus of TFPR-T. SWISS-MODEL analysis of TFPR-hd and TFPR-ta revealed that the conformational structures of TFPR-hd and TFPR-ta are relative intact with different extent of disruption of α-β-α sandwich. Lacking of β4-fold (TFPR-hd) led to β2 and β3-fold disappearing at the same time just as TFPR-T, but due to the intactness of α1-helix, the intact two outer layers: two α-helices at N-terminus, and one α-helix at C-terminus still retained; lacking of partial α1-helix at the N-terminus (TFPR-ta) did not affect the integrity of the unique α-β-α sandwich conformation, suggesting that these two fragments might retain immune regulatory activity and adjuvanticity. As expected, dendritic cells stimulating assay in vitro and adjuvant evaluation assay in BALB/c mice using OVA demonstrated that both TFPR-hd and TFPR-ta have similar activity to activate dendritic cells and adjuvanticity to TFPR1, indicating that the integrity of conformational structure is essential for proteins to maintain their immunologic activity [49,50,51], α1-helix and β4-fold should not be removed at the same time. The secondary structural analysis by CD-spectrum preliminarily proved the conformational changes of different fragments; in order to better interpret the relationship between the conformation and function, the more accurate conformational structures of these proteins need to be dissolved by other more precise method, such as nuclear magnetic resonance (NMR) [52] and X-ray [53] diffraction in the future.

On the other hand, because of the similar high immunogenicity of TFPR-hd as TFPR1, and significantly reduction of immunogenicity of TFPR-ta due to the removal of a B cell dominant epitope (P1), TFPR-ta is more suitable to be applied as an adjuvant. However, TFPR-ta still has some extent immunogenicity, to obtain a functional fragment with much lower immunogenicity, we will identify more B epitopes such as a predicted strong B epitope #4 (SHSSRDSRV), and attempt to remove more B cell epitopes located in the middle part of the protein by mutation [54,55,56,57].

## 5. Conclusions

We not only proved that integrity of an α-β-α sandwich conformation is essential for TFPR1 to maintain its immunologic activity and adjuvanticity; but also obtained a functional truncated fragment TFPR-ta ranging from 40–168 Aa of triflin, which has similar adjuvanticity to TFPR1 but with 2-log fold lower immunogenicity. Also, we set up a practical method to obtain a functional fragment but with minimum immunogenicity for optimization of protein-based adjuvant and other protein-based therapeutics. This functional TFPR-ta with less immunogenicity has potential to be developed into a novel vaccine adjuvant, especially for peptide-based vaccine.

## Figures and Tables

**Figure 1 biomolecules-09-00869-f001:**
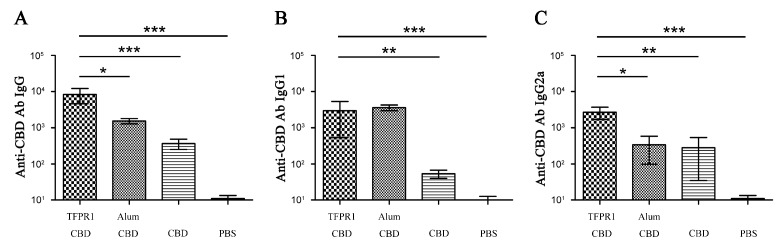
Detection of anti-CBD antibodies in BALB/c mice immunized with TFPR1 and CBD. Female 6- to 8-week-old BALB/c mice were immunized intramuscularly with synthetic HIV-1 CBD (20 μg) plus recombinant TFPR1 (10 μg), and boosted once three weeks later. Mice were immunized with HIV-1 CBD plus alum (volume ratio of OVA to alum is 1:9), HIV-1 CBD alone and PBS as control. Serum was collected on the second week after the final immunization, and anti-CBD-specific antibodies IgG (**A**), IgG1 (**B**) and IgG2a (**C**) were measured by ELISA which coated with HIV-1 CBD (final concentration, 1 μg/mL). The antibody titer was expressed as the reciprocal of the largest dilution of serum which OD_450_ value was greater than average OD value of negative serum plus two-fold of SD. One-way ANOVA was used for statistical analysis.* denotes *p*<0.05, ** *p* < 0.01, *** *p* < 0.001.

**Figure 2 biomolecules-09-00869-f002:**
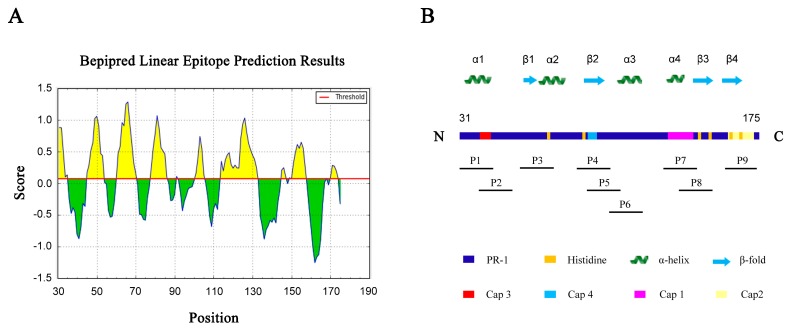
Prediction of the B cell dominant epitopes on TFPR1. (**A**) The potential B cell epitopes on TFPR1 were predicted and analyzed using the online software IEDB. (**B**) Schematic showing the structural and functional motifs of TFPR1, including the pathogenesis-related protein 1 (PR-1), histidine, α-helix (green), and β-fold (blue) according to the online software NPS (http://npsa-prabi.ibcp.fr) by the tool “secondary structure consensus prediction” and the reference [24], and location on TFPR1 of nine peptides (P1–P9, Table 2) which are both key motifs and predicted B cell epitopes were shown under the schematic diagram.

**Figure 3 biomolecules-09-00869-f003:**
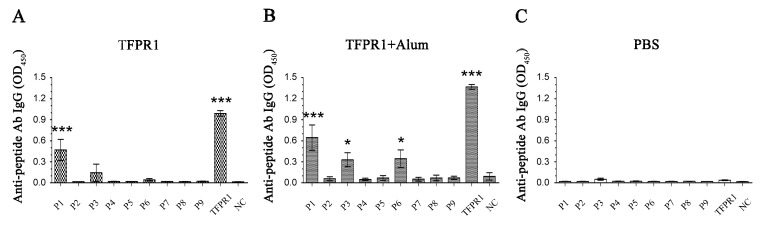
Identification of the B cell dominant epitopes on TFPR1. Female 6- to 8-week-old BALB/c mice were immunized intramuscularly with TFPR1 (10 μg) alone (**A**) or TFPR1 plus alum (volume ratio of OVA to alum is 1:9) (**B**), endotoxin-free PBS alone (**C**) was used as the control; and boosted once three weeks later. Serum was collected on the second week after the final immunization, and anti-peptide-specific antibodies IgG were measured by ELISA which coated with each synthetic peptide, P1–P9 (final concentration, 2 μg/mL), and TFPR1 as positive control. Data was presented as the OD_450_ value at 1/20 dilution. One-way ANOVA was used for statistical analysis. * denotes *p* < 0.05, *** *p* < 0.001.

**Figure 4 biomolecules-09-00869-f004:**
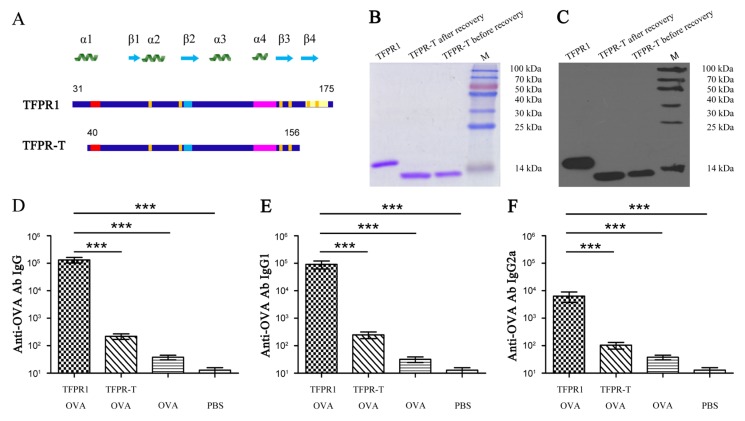
Schematic presentation and expression in *Escherichia coli*. of TFPR-T and its adjuvant activity in BALB/c mice immunized with OVA. (**A**) Schematic showing the structural and functional motifs on TFPR-T compared to TFPR1. (**B**) SDS-PAGE analysis of TFPR-T after purification and recovery. (**C**) Identification of TFPR-T after purification and recovery by Western blotting using mouse sera containing anti-TFPR1 antibody. (**D**) Titer of anti-OVA antibody IgG, (**E**) titer of anti-OVA antibody IgG1, and (**F**) titer of anti-OVA antibody IgG2a. One-way ANOVA was used for statistical analysis, *** *p* < 0.001.

**Figure 5 biomolecules-09-00869-f005:**
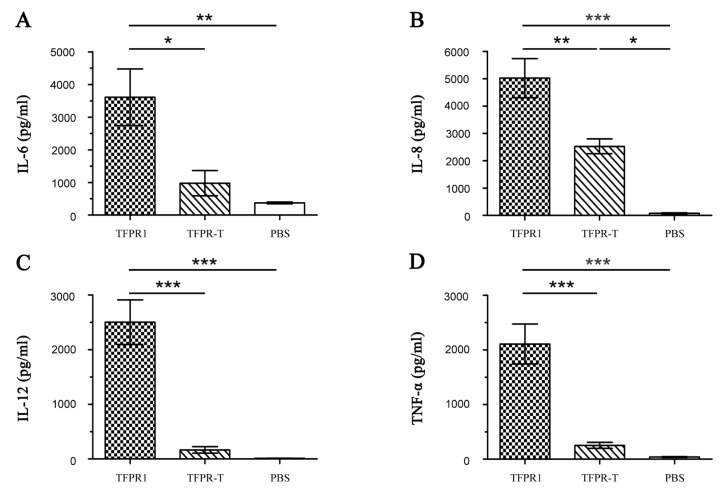
Detection of cytokine levels secreted by dendritic cells incubated with TFPR-T for 24 h. Bone marrow cells were collected from 4- to 5-week-old healthy male BALB/c mice under the aseptic conditions; and were cultured with complete RPMI 1640 containing rmGM-CSF (final concentration, 10 ng/mL) and rmIL-4 (final concentration, 10 ng/mL) for six days. TFPR-T (final concentration, 10 μg/mL) was added on day six, and cells were incubated for another 24 h, TFPR1 (final concentration, 10 μg/mL) was used as positive control, and PBS as negative. Culture supernatants were collected on day seven, and the levels of IL-6 (**A**), IL-8 (**B**), IL-12 (**C**) and TNF-α (**D**) were measured using specific ELISA kits. Cytokine concentration was read from the standard curves and expressed as pg/mL. One-way ANOVA was used for statistical analysis. * denotes *p* < 0.05, ** *p* < 0.01, *** *p* < 0.001.

**Figure 6 biomolecules-09-00869-f006:**
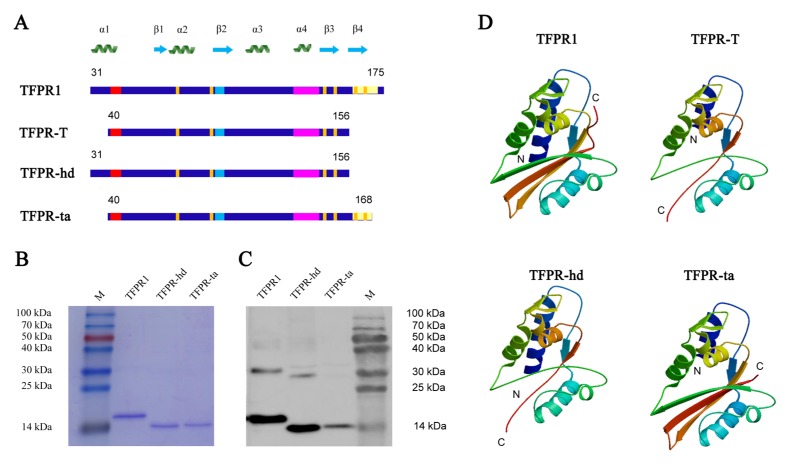
Schematic secondary and predicted conformational structure, and expression in *E. coli*. of TFPR-hd and TFPR-ta. (**A**) Schematic showing the structural and functional motifs of TFPR-hd and TFPR-ta compared to TFPR1. (**B**) SDS-PAGE analysis of TFPR-hd and TFPR-ta after purification and recovery. (**C**) Identification of TFPR-hd and TFPR-ta after purification and recovery by Western blotting using mouse sera containing anti-TFPR1 antibody. (**D**) The conformational structures of TFPR1, TFPR-T, TFPR-hd, and TFPR-ta were predicted by SWISS-MODEL.

**Figure 7 biomolecules-09-00869-f007:**
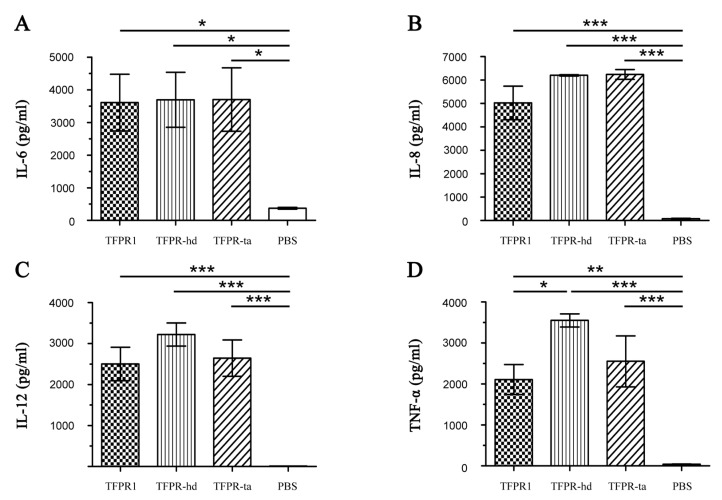
Detection of cytokine levels secreted by dendritic cells incubated with TFPR-hd or TFPR-ta for 24 h. Bone marrow cells were collected from 4- to 5-week-old healthy male BALB/c mice under the aseptic conditions; and were cultured with complete RPMI 1640 containing rmGM-CSF (10 ng/mL) and rmIL-4 (10 ng/mL) for six days. TFPR-hd and TFPR-ta (10 μg/mL) were separately added on day six, and cells were incubated for another 24 h, TFPR1(10 μg/mL) was used as positive control, and PBS as negative. Culture supernatants were collected on day seven, and the levels of IL-6 (**A**), IL-8 (**B**), IL-12 (**C**) and TNF-α (**D**) were measured using specific ELISA kits. Cytokine concentration was read from the standard curves and expressed as pg/mL. One-way ANOVA was used for statistical analysis, * denotes *p*< 0.05, ** *p* < 0.01, *** *p* < 0.001.

**Figure 8 biomolecules-09-00869-f008:**
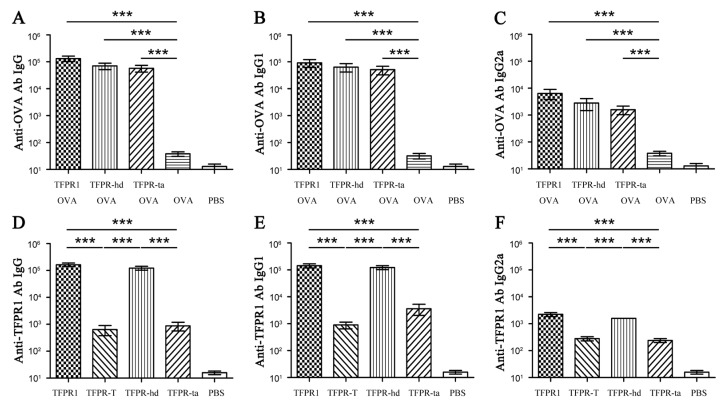
Detection of anti-OVA and anti-TFPR1 antibodies in BALB/c mice immunized with OVA and TFPR-hd or TFPR-ta. Female 6- to 8-week-old BALB/c mice were immunized intramuscularly with endotoxin-free OVA (10 μg) plus recombinant TFPR-hd or TFPR-ta (10 μg), and boosted once three weeks later. Mice were immunized with OVA plus TFPR1, OVA alone and PBS as control. Serum was collected on the second week after the final immunization, and anti-OVA-specific antibodies IgG (**A**), IgG1 (**B**) and IgG2a (**C**) were measured by ELISA which coated with OVA (final concentration, 1 μg/mL), meanwhile, anti-TFPR1-specific antibodies IgG (**D**), IgG1 (**E**), and IgG2a (**F**) were measured by ELISA which coated with TFPR1 (final concentration, 1 μg/mL). The antibody titer was defined as the reciprocal of the largest dilution of serum which OD_450_ value was greater than OD value of negative serum plus two-fold of SD. One-way ANOVA was used for statistical analysis, denotes *p* < 0.001.

**Table 1 biomolecules-09-00869-t001:** Prediction of B cell epitopes on TFPR1 using online IEDB software *.

No.	Start	End	Amino Acid Sequence	Length (Aa)	Score	Secondary Structures ^#^
1	31	35	PEIQN	5	0.882	α1-helix
2	45	53	RRSVNPTAS	9	1.06	α1-helix
3	61	70	YPEAAANAER	10	1.287	β1-fold, α2-helix
4	78	86	SHSSRDSRV	9	1.07	/
5	101	105	YPAKW	5	0.741	α3-helix
6	114	132	GEYKDFKYGVGAVPSDAVI	19	1.033	α3-helix
7	145	147	RAG	3	0.246	β3-fold
8	149	157	AAAYCPSSK	9	0.649	β3-fold
9	170	174	GNIIG	5	0.283	/

* IEDB website (http://tools.iedb.org/main/bcell/); ^#^ The amino acid sequences of these B cell epitopes partially overlap with the corresponding secondary structure sequences.

**Table 2 biomolecules-09-00869-t002:** Information of nine peptides on TFPR1 for B cell dominant epitope screening.

Peptide	Start	End	Amino Acid Sequence	Length (Aa)	Predicted B Epitope ^#^	Secondary Structures
P1	31	48	PEIQNEIIDLHNSLRRSV	18	high(#1+2)	α1-helix *
P2	41	55	HNSLRRSVNPTASNM	15	high(#2)	α1-helix
P3	57	71	KMEWYPEAAANAERW	15	high(#3)	β1-fold *, α2-helix
P4	89	103	GIKCGENIYMATYPA	15	N.P.(#5^Δ^)	β2-fold *
P5	94	108	ENIYMATYPAKWTDI	15	intermediate(#5)	β2-fold, α3-helix
P6	103	117	AKWTDIIHAWHGEYK	15	intermediate(#5+6)	α3-helix *
P7	132	146	IGHYTQIVWYKSYRA	15	N.P	α4-helix *, β3-fold
P8	138	153	IVWYKSYRAGCAAAYC	16	low(#7)	β3-fold*
P9	158	172	YSYFYVCQYCPAGNI	15	low(#9^Δ^)	β4-fold*

Note: (1) “high” means the peptide contains predicted B epitope which score ≥ 1, “intermediate” stands for 0.5 ≤ score < 1, “low” stands for 0.2 ≤ score < 0.5, “N.P” stands for contains no predicted B epitopes. Δ stands for this peptide contains only 2–3 Aa of the predicted epitope. (2) ^#^ Epitope number in Table 1, * the amino acid sequences of the peptide contain the entire corresponding secondary structure sequences, while the others which are not labeled with asteroid contain part of the secondary structure.

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
