# Peer review of "The Integrity of α-β-α Sandwich Conformation Is Essential for a Novel Adjuvant TFPR1 to Maintain Its Adjuvanticity"

_biomolecules, 2019, doi:10.3390/biom9120869_

Round 1
Reviewer 1 Report
The manuscript "The integrity of α-β-α sandwich conformational structure is essential for a protein-based adjuvant TFPR1 to maintain its immunologic activity and adjuvanticity" displays in a clarify manner a novel approach to evaluate and improve the activity of protein-based vaccine adjuvant throught a good rapresentation and description of the data. The aim of the study is interesting and in my opinion the manuscript can be accepted with a minor revision.
I suggest:
a moderate english language revision. In addition, to permit a better understand the novelty of the paper and to increase the important of this study, I suggest to increase the description of the relevant advantages that this discovery permit, and add these in discussion or conclusion session.
Reviewer 2 Report
New protein-based vaccine adjuvants such as TFPR1, and derivatives thereof, may contribute to novel or improved vaccines. It's important to study the mechanism of its adjuvanticity. The information can be used to engineer the protein to maximize its potency or improve its safety. It is also essential for designing quality control assays for the new adjuvant.
The authors previously have published on adjuvanticity of TFPR1 in Vaccine Sun et al. , 2019). It the new manuscript, here further reviewed, the authors made an attempt to improve TFPR1 as a vaccine adjuvant for peptide vaccines. I think this research is valuable, however, I have multiple concerns about this article.
The title: I do not think the sandwich is essential for its immunologic activity. Suggestion: The integrity of the alfa-beta-alfa sandwich conformation is essential for TFPR1 to maintain its adjuvanticity.
The language: The authors need to improve the language of the manuscript. Some findings (not limited): the abstract. The second sentence is way too long and grammatically incorrect. It's also B cell epitopes. Line 21: delete significantly. Lines 29-30: rewrite the first sentence of the introduction. line 51 replace relevant with related. line 136: antibodies. line 155: alum. line 160: but retaining ... may be important. line 199: Strong P-1-specific antibody response. line 202: delete space between in and mice. line 203: rephrase: the most predicted epitopes with high scores are indeed epitopes.
References: Your previous data, which you mention in multiple places, is not referenced.
Methods: It is unclear to me how you selected the nine peptides from table 2. line 185: NPS@, What tool did you use from that server?
Conclusions: line 203: I do not agree. Many predicted epitopes are not, or only partially represented in the nine tested peptides. What about epitopes 4 and 6?
line 222: I see no data indicating that TFPR-T was in its native form. I see data including SDS page and Western Blot, but in these type of assays, all structure is basically lost. This is the biggest caveat. The authors claim the structure is crucial, but no biophysical data (Circular Dichroism or FTIR) is shown to demonstrate that these structures are indeed present after refolding. I think this is needed.
You claim that the use of dendritic cells to demonstrate the adjuvanticity of the proteins is a novel approach. It is a valid model, but this has been done by others and therefore I believe you should reword your conclusions.
TFPR-ta has equal adjuvanticity and 2-log fold reduced immunogenicity because the structure was maintained but the first epitope deleted. Other epitopes cannot be deleted because it would disturb the structure. Did the authors try mutations instead of deletions? see literature: removal of B cell epitopes as a practical approach for reducing the immunogenicity of foreign protein-based therapeutics Satoshi Nagataa and Ira Pastanb Adv Drug Deliv Rev. 2009.
Round 2
Reviewer 2 Report
Important structural data was included and the manuscript was significantly improved. I wonder how much noise there was on the CD data when measuring at low UV using PBS (Cl- ions usually give a high background signal which may saturate the detector).
